# SIRE: SE(3) Intrinsic Rigidity Embeddings

**Cameron Smith**[1]    **Basile Van Hoorick**[2]    **Chonghyuk Song**[3]
**Vincent Sitzmann**[3]    **Vitor Guizilini**[2*]    **Yue Wang**[1*]

[1]**University of Southern California**    [2]**Toyota Research Institute**
[3]**Massachusetts Institute of Technology**    ***Equal Advising***

Reviewed on OpenReview: https://openreview.net/forum?id=OZ9HOTOYMt

## Abstract

Motion serves as a powerful cue for scene perception and understanding by separating independently moving surfaces and organizing the physical world into distinct entities. We introduce **SIRE**, a self-supervised video-learning formulation that learns intrinsic rigidity embeddings to drive soft motion-based segmentation and rigid-body reasoning. Rigidity embeddings softly group pixels into rigid components, where pixels belonging to the same rigid object share similar embeddings. The core of our formulation is a simple point-tracking reconstruction loss: our model estimates rigidity embeddings, which is used to lift off-the-shelf 2D point track trajectories into rigid SE(3) tracks, which are simply re-projected back to 2D and compared against the original 2D tracks for supervision. Crucially, our framework is fully end-to-end differentiable and can be optimized either on video datasets to learn generalizable image-encoder priors, or even on a single video to capture scene-specific structure — highlighting strong data efficiency. We demonstrate the effectiveness of our rigidity embeddings across multiple tasks, and our findings suggest that SIRE can learn powerful geometry and motion rigidity priors from video data with minimal supervision.

## 1 Introduction

Motion is a powerful cue for organizing the structure of the physical world. In cognitive neuroscience, empirical studies support the Gestalt principle of *common fate* (Wertheimer, 1938; Spelke, 1990; Braddick, 1993), which refers to the tendency of the human visual system to group together elements based on synchronized movement. Prior works have been inspired by these foundational ideas and leveraged motion-based cues for learning object segmentation. Representative methods often structure their model to explain a video's optical flow using a set of unsupervised object masks (Elsayed et al., 2022; Kipf et al., 2022; Yang et al., 2021). However, these works almost exclusively operate without 3D awareness and are therefore limited to synthetic scenes or videos with simple motion.

In parallel, recent methods have emerged for self-supervised learning of geometry and camera pose estimation from the same off-the-shelf optical flow supervision signal (Smith et al., 2024; 2023; Bowen et al., 2022). These methods are 3D-aware and operate on real-world scenes with more complex motion, but are generally limited to static scenes: they operate by explaining the optical flow via a single SE(3) transformation — namely, the camera motion. But dynamic videos importantly contain not just a *single* rigid body transformation; instead, there are a potentially unbounded number of them, in which the camera and dynamic scene content move independently.

We propose SE(3) Intrinsic Rigidity Embeddings (SIRE), which effectively combines these two lines of work to learn object-aware embeddings on real-world scenes featuring diverse motion. SIRE is 3D-aware and can be learned from casual real-world videos. Our first innovation is to learn the underlying scene dynamics via SIRE, a representation that softly encodes the scene's distinct rigid groups, and second is a simple 4D reconstruction loss which enables training these embeddings without direct supervision. A high-level overview is shown in Fig. 1.

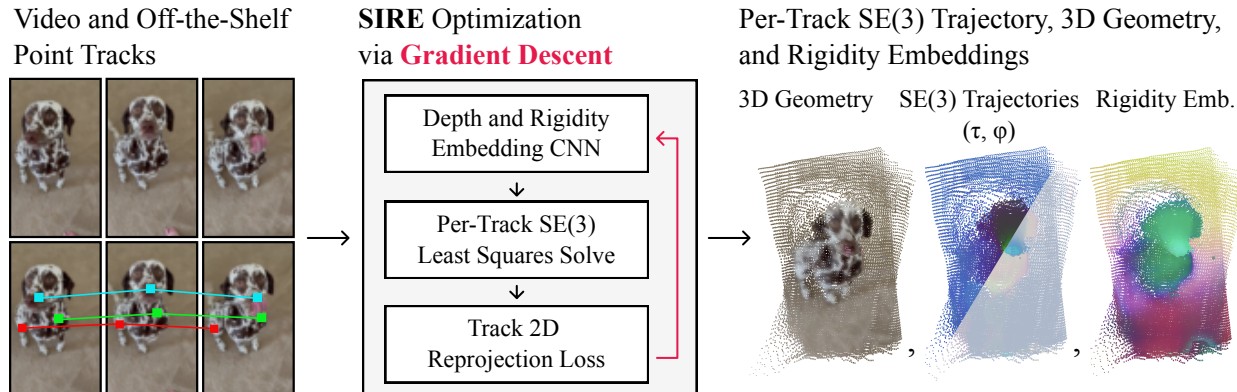

Figure 1: **SIRE** learns intrinsic rigidity embeddings—features that encode rigid-object priors—directly from raw video and off-the-shelf 2D point tracks. An image encoder predicts per-pixel embeddings that softly group pixels into rigid bodies. A lightweight SE(3) solver lifts each 2D track into an SE(3) rigid-body trajectory; those trajectories are re-projected to 2D and compared against the original tracks for supervision. Because the entire pipeline is end-to-end differentiable, SIRE can be trained on large video collections to learn generalizable motion-aware priors, or even fine-tuned on a single clip for scene-specific features.

We conduct extensive experiments on broad tasks to validate our model's learned rigidity embeddings as well as intermediate geometry representations. Our experiments demonstrate that both dataset-wide and even per-video optimization of our rigidity embeddings produces strong semantic features useful for downstream segmentation tasks. Our findings suggest that this simple formulation can pave the way towards self-supervised learning of priors over geometry and object rigidities from large-scale video data.

## 2 Related Work

### 2.1 Self-Supervised Motion-Based 2D Segmentation Learning

Using optical flow as a supervision signal for learning meaningful object features is promising: it naturally captures motion boundaries and can reveal distinct object structure by grouping regions that move rigidly together, without relying on explicit object labels. Understanding and decomposing scene motion from optical flow video has been widely studied in self-supervised settings. Most works in this line leverage slot-based model architectures which effectively explain the motion using $N$ object groupings. SAVi (Kipf et al., 2022) learns compelling object-centric decompositions, but is empirically limited to simple synthetic scenes. Self-Supervised Video Object Segmentation by Motion Grouping (Yang et al., 2021) operates on real-world videos, but uses slot attention with just two slots, effectively achieving foreground-background segmentation, and empirically requires the background flow to be planar (via rotation-dominant camera captures).

These methods are not 3D-aware, meaning they cannot leverage full 3D scene structure: consider that the optical flow induced by a camera translating even through a static scene induces complex 2D motion due to perspective effects, but in 3D it is explained by a single SE(3) motion. In contrast, our approach is 3D-aware, allowing it to generalize beyond videos with simple scene motion. Importantly, our method does not assume a fixed number of rigid components or 'slots'. We demonstrate that our approach can produce compelling multi-body soft-segmentations, even in scenes with complex, non-rigid motions.

### 2.2 Self-Supervised Structure-from-Motion (SfM) Methods

Another class of approaches (Smith et al., 2023; 2024) use the same off-the-shelf optical flow and point tracks (Teed & Deng, 2020; Xu et al., 2022; Karaev et al., 2023; Doersch et al., 2023) as supervision, but leverage 3D-aware formulations and aim to learn compelling geometry and camera pose estimation from monocular videos of static scenes. FlowCam (Smith et al., 2023) and FlowMap (Smith et al., 2024) both demonstrate that it is possible to train self-supervised depth and camera pose estimation just from optical

**1) Per-Frame Network Estimates Depth and Rigidity Embedding**

**2) Per-Track Rigidity-Weighted SE(3) Solving**
2(a): Lift 2D tracks into 3D Scene Flow   2(b): Extract Track Rigidities   2(c): Solve Per-Track SE(3)

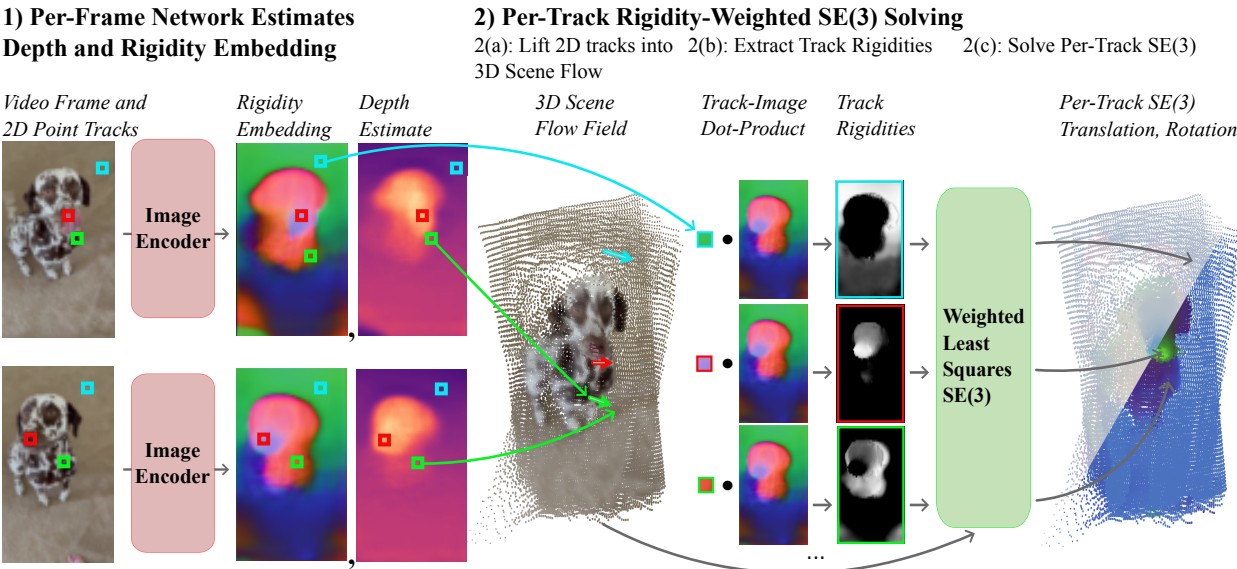

Figure 2: **A SIRE training forward pass.** Given two frames and 2D point tracks, a per-frame CNN first estimates rigidity embeddings as well as depth. We lift the 2D point tracks into 3D scene flow via the depth estimates. For each point track, we extract a rigidity map by comparing its rigidity embedding to all other point track rigidity embeddings, then solve for the SE(3) transformation on the global scene flow using the rigidity map as confidence weights in the solver. Lastly, the per-track SE(3) trajectories are reprojected back to 2D and compared with the original 2D trajectory for supervision.

flow supervision. They introduce a simple formulation for geometry and camera pose estimation by lifting the estimated depth and off-the-shelf 2D flow into 3D flow and *solving* for the camera pose; then, they supervise their estimation by comparing the off-the-shelf 2D flow to that induced by the predicted depth and camera motion.

These methods have delivered promising results but are critically restricted to modeling static scenes, where the observed optical flow can be explained by a single SE(3) motion (the camera pose). Our method builds upon FlowMap to inherit its simple formulation and strong performance, but we extend it to model dynamic scenes. At a high level, we extend it by solving for an SE(3) motion per 2D point track, instead of one global SE(3) camera transformation. This enables us to train on unconstrained, unlabeled videos featuring non-rigid dynamics.

Other works have attempted to approximate the classical SfM pipeline (Schönberger & Frahm, 2016; Mur-Artal et al., 2015; Mur-Artal & Tardós, 2017; Campos et al., 2021; Rosinol et al., 2020) with deep learning counterparts, but these methods typically either have neural networks that directly estimate camera poses and are less accurate (Tang & Tan, 2018; Czarnowski et al., 2020; Zhou et al., 2018; Clark et al., 2018; Ummenhofer et al., 2017; Liu et al., 2019; Teed & Deng, 2018; Wang et al., 2021b; Bloesch et al., 2018), or are non-differentiable and approximate only subsets of the pipeline (Choy et al., 2016; Mishchuk et al., 2017; Luo et al., 2018; Ono et al., 2018; Sarlin et al., 2020; Zhao et al., 2022; Harley et al., 2022; Doersch et al., 2023).

## 2.3 Intrinsic Rigidity Embeddings

Intrinsic rigidity embeddings softly group points into rigid bodies: two points with similar embeddings are interpreted as belonging to the same rigid group. They were first introduced in RAFT-3D (Teed & Deng, 2021) to estimate piecewise-constant SE(3) maps for higher quality scene flow prediction. These embeddings are compelling because they a) avoid deciding how many rigid bodies are in the scene a-priori, thus supporting a potentially unbounded number of rigid bodies; and b) offer soft attention-like gradients via their dot-product response formulation.

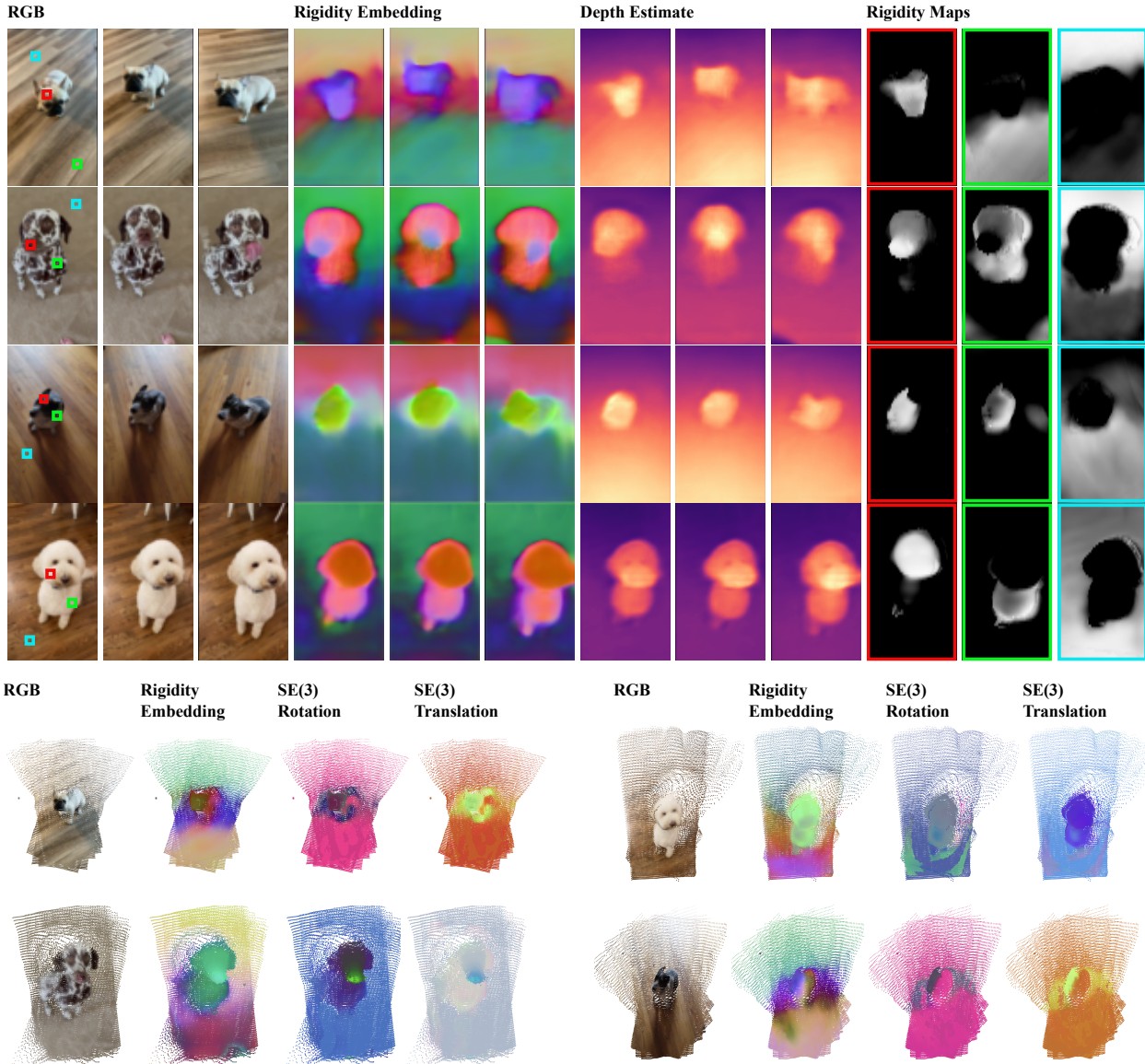

Figure 3: **Results on the CO3D-Dogs Dataset.** Here we plot estimated rigidity embeddings, intermediate depth estimates, and highlighted rigidity maps (top row). And while geometry estimation is not the primary goal of our method, we also plot accumulated 4D point clouds and color-coded SE(3) rotation and translation components (bottom row). Observe how SE(3) components within rigid bodies are often constant. Note that rigidity maps are estimated per-track and we manually pick a few representative ones here; consider how they yield semantically meaningful soft segmentations without supervision.

However, a main limitation of this original implementation is that ground-truth scene flow is required to train, which makes its usage largely restricted to simulated data. In this work, we instead show how intrinsic rigidity embeddings can be learned directly from video data using just off-the-shelf 2D flow.

## 2.4 Dynamic Scene Reconstruction

Until recently, reconstructing dynamic scenes was considered an outstanding task in computer vision. Enabled by recent breakthroughs in monocular depth and point-track estimation, some methods (Wang et al., 2024; Lei et al., 2024) have demonstrated strong 4D reconstructions on challenging dynamic scenes. Older works also leveraged monocular depth estimates and neural rendering (Kerbl et al., 2023; Mildenhall et al., 2020) but were less robust, as were the depth and point track estimators they depended on (Li et al., 2019;

| Method | Bear | Blackswan | Camel | Cows | Elephant | Flamingo | Goat | Lucia |
|---|---|---|---|---|---|---|---|---|
| Ours-Rig.Emb | 0.834 | 0.590 | 0.729 | 0.818 | 0.753 | 0.745 | 0.798 | 0.670 |
| Ours-Enc. | 0.870 | 0.807 | 0.814 | 0.837 | 0.767 | 0.774 | 0.790 | 0.643 |
| ImageNet Enc. | 0.659 | 0.179 | 0.103 | 0.123 | 0 | 0.253 | 0.083 | 0.217 |
| DINOv2† | 0.944 | 0.930 | 0.915 | 0.923 | 0.917 | 0.797 | 0.885 | 0.861 |

Table 1: **Downstream Segmentation Results.** We provide quantitative comparisons (IOU; higher is better) for downstream moving-object segmentation on several DAVIS scenes. We threshold the segmentation predictions at .5, resulting in a zero score for the 'ImageNet Enc.' baseline on the Elephant scene.  †Note that DINOv2 features are trained on large image datasets, whereas our results here are generated *per-video*.

Zhang et al., 2022; Bansal et al., 2020; Fridovich-Keil et al., 2023; Broxton et al., 2020; Li et al., 2022b; Cao & Johnson, 2023; Song et al., 2023; Wang et al., 2022; Li et al., 2024a; Işık et al., 2023; Li et al., 2022a; Weng et al., 2022; Du et al., 2021; Pumarola et al., 2021; Park et al., 2021a; Wang et al., 2021a; Park et al., 2021b; Xian et al., 2021; Li et al., 2023; Wu et al., 2024; Duan et al., 2024; Yang et al., 2023; 2024; Gao et al., 2022). These methods typically segment the scene into static background and dynamic foreground regions, reconstruct the background using classical SfM approaches, then re-integrate the foreground via re-rendering losses.

While these methods' results are indeed impressive, they are importantly non-differentiable due to their multi-stage reconstructions and therefore can not be embedded in machine learning pipelines to learn generalizable priors over video datasets. Although our aims are different, note that our method is simple and fully differentiable, allowing us to train on a *dataset* of videos to learn generalizable priors, paving the path towards large-scale video learning.

## 3 SE(3) Intrinsic Rigidity Embeddings

Our method, dubbed *SE(3) Intrinsic Rigidity Embeddings* (SIRE), takes in a dataset of videos with corresponding 2D point tracks and trains an image encoder to learn generalizable priors over object rigidity.

At a high level, our method trains a per-frame image encoder to predict $M$-dimensional rigidity embeddings $F \in \mathbb{R}^{H \times W \times M}$ as well as depth maps $D \in \mathbb{R}^{H \times W \times 1}$ (though note that our aim is not to learn highly accurate geometry; geometry is just an intermediate property in our formulation). Given a video with $T$ frames and $N$ point tracks, our training forward pass lifts the 2D point tracks $p \in \mathbb{R}^{N \times T \times 2}$ into SE(3) trajectories $P \in \mathbb{R}^{N \times T \times 6}$ via a least-squares solve using the estimated depth and rigidities, projects them back to 2D trajectories, and minimizes the difference with their original locations to supervise the depth and rigidity networks. We begin by describing how we can solve for a *global* SE(3) trajectory (temporarily assuming a static scene) via the FlowMap formulation, then extend it to modeling dynamic scenes by solving for a potentially unique SE(3) trajectory per point track, and conclude with our complete formulation.

### 3.1 Solving for Global SE(3) Trajectories

Given a video with corresponding 2D point tracks $p$, our first goal is to lift these 2D trajectories into SE(3) trajectories $P$. If we assume temporarily that there is only one rigid body in the scene (i.e. the scene is static), FlowMap (Smith et al., 2024) showed that we can actually *solve* for the rigid transformation by lifting the 2D tracks into 3D scene flow via a depth estimate $D$, then solving for the SE(3) transformation which best explains the scene flow. Specifically, to solve for the SE(3) transformation $P_{it}$ for point track $i$ from time $t$ to $t_{+1}$, they use the Procrustes formulation $\underset{P_{it} \in \text{SE(3)}}{\text{argmin}} \left\| W \left( d_{*t} K^{-1} p_{*t} - P_{it} d_{*t_{+1}} K^{-1} p_{*t_{+1}} \right) \right\|_2^2$, where $p_{*t}$ refers to all point tracks at time $t$, $d_{it}$ refers to the depth of point track $i$ at time $t$, and $W$ is a diagonal confidence matrix to down-weight bad correspondences (e.g. from incorrect flow in sky regions). This weighted least-squares problem is differentiable and can be solved via singular value decomposition (Choy et al., 2020; Smith et al., 2023).

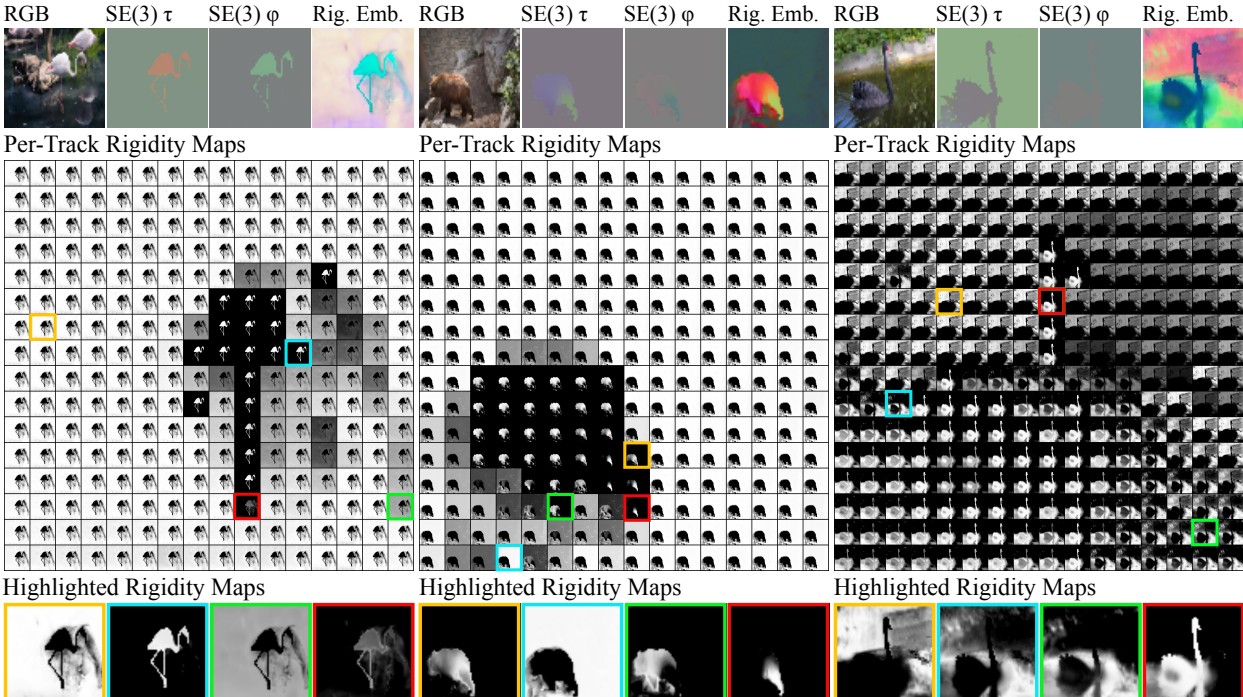

Figure 4: **Rigidity Response Grid.** Here we plot, for three scenes, the rigidity response grids – where each cell contains a rigidity map from the point track at that location to all other point tracks. While we only plot a 16x16 grid here for space constraints, note we in practice use 64x64 grids of point tracks. We also include (top) the per-track images of RGB, SE(3) rotation (Euler angles) $\phi$ and translation vector $\tau$, and rigidity embeddings, and (bottom) manually highlighted rigidity maps on the bottom row. The center example (bear) clearly shows that the largest component (blue) corresponds to the background (camera movement), while point tracks on the bear's leg (red) form another distinct group (right leg movement). Note that these are results of per-scene optimizations without depth supervision.

## 3.2 Solving for Local SE(3) Trajectories

Dynamic scenes evidently feature more than just the camera motion — consider a scene with falling snow or running water, where each observed point track might be induced by a unique underlying rigid body; solving for a global SE(3) trajectory is no longer appropriate. Assume for simplicity that we know which points belong to the same rigid bodies; that is, consider that for each point track $p_i$ we have a binary mask $R_i$ over all other point tracks $p_j$: $R_i = \{\text{rig}(p_i, p_j) | \forall p_j\}$ where $\text{rig}(p_i, p_j)$ is 1 if points $p_i$ and $p_j$ belong to the same rigid group and 0 otherwise.

We can use $R_i$ to solve for $P_{it}$ by ignoring all the scene flow vectors not in the same rigid group, multiplying the confidence weights by the per-track rigidity weights $R_i$:

$$\underset{\mathbf{P}_{it} \in \text{SE}(3)}{\text{argmin}} \left\| R_i W \left( d_{*t} K^{-1} p_{*t} - P_{it} d_{*t_{+1}} K^{-1} p_{*t_{+1}} \right) \right\|_2^2 \tag{1}$$

## 3.3 Softly Parameterizing Per-Track Rigidity Masks via Intrinsic Rigidity Embeddings

Assuming known rigidity masks is impractical outside of simulation; instead, we need to estimate them. One simple and efficient way to do is via rigidity embeddings (introduced by RAFT-3D (Teed & Deng, 2021)), a pixel-aligned $M$-dimensional feature map where two points are in same rigid body if they have similar embeddings. Formally, given two tracks $p_i$ and $p_j$ and corresponding embedding features $f_i$ and $f_j$, their rigidity is parameterized as their cosine similarity, lower-bounded at 0. We can now define the rigidity mask

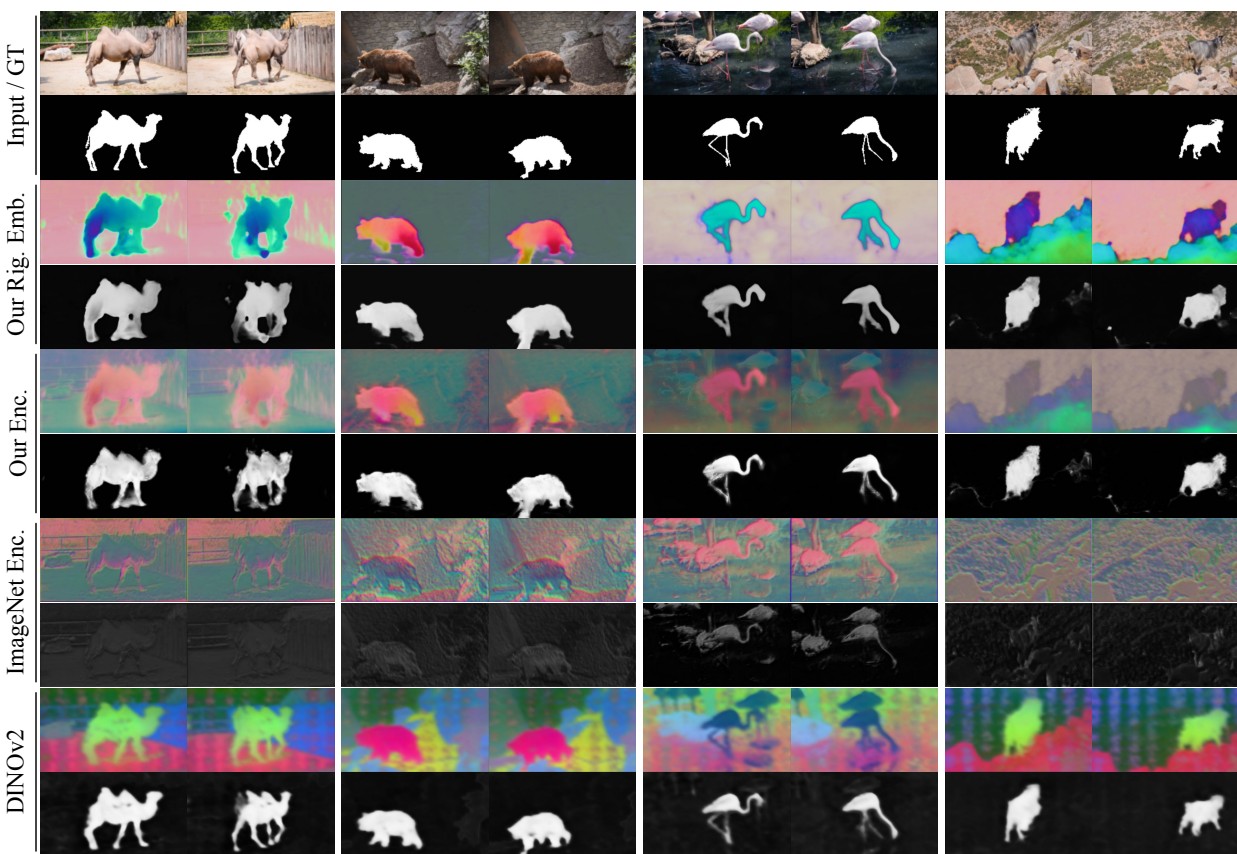

Figure 5: **Downstream Segmentation Plots.** We demonstrate that our method's embeddings are useful for downstream moving object segmentation by freezing features from our model and baselines and training a two-layer MLP for segmentation. We compare using features from our rigidity embeddings, the last layer of our trained feature backbone, and the feature backbone before our training. Note these embeddings are trained from single-scene optimizations on each of these videos and without depth supervision. For each method, we visualize the PCA of their features (top) and their segmentations (bottom).

for $p_i$ as:

$$R_i = \left\{ max \left( 0, \frac{f_i \cdot f_j}{\|f_i\|\|f_j\|} \right) |\forall f_j \right\} \tag{2}$$

### 3.4 Supervision via SE(3) Induced 2D Point Tracks

One problem remains: while we can robustly estimate 2D point tracks for in-the-wild videos, we do not necessarily have corresponding *ground-truth* SE(3) trajectories, rigidity masks, and perhaps not even depth estimates, to train our model. As illustrated in Fig. 2, to supervise our model's predictions, we lift each point track in frame $t$ to 3D via the depth estimate, transport the point in 3D to the next frame $t + 1$ via the estimated SE(3) transformation, project back to 2D, and minimize the difference:

$$\left\| K P_{it} d_{it} K^{-1} p_{it} - p_{it_{+1}} \right\|_2^2 \tag{3}$$

### 3.5 Full SIRE Forward Pass

Given a video with intrinsics $\mathbf{K}$ and 2D point tracks $\{\mathbf{p}_{it}\}$, the shared image encoder predicts per-frame depth $D_t$ and rigidity embeddings $F_t$ (Sec. 3). For each track $i$ and adjacent pair $(t, t+1)$, we unproject the 2D points to 3D using $D_t$ and $\mathbf{K}$ and solve a rigidity-weighted Procrustes problem to obtain the relative motion $\mathbf{P}_{i,t} \in SE(3)$ via Eq. equation 1. *Frame of reference:* Note that depths are defined in each camera-$t$

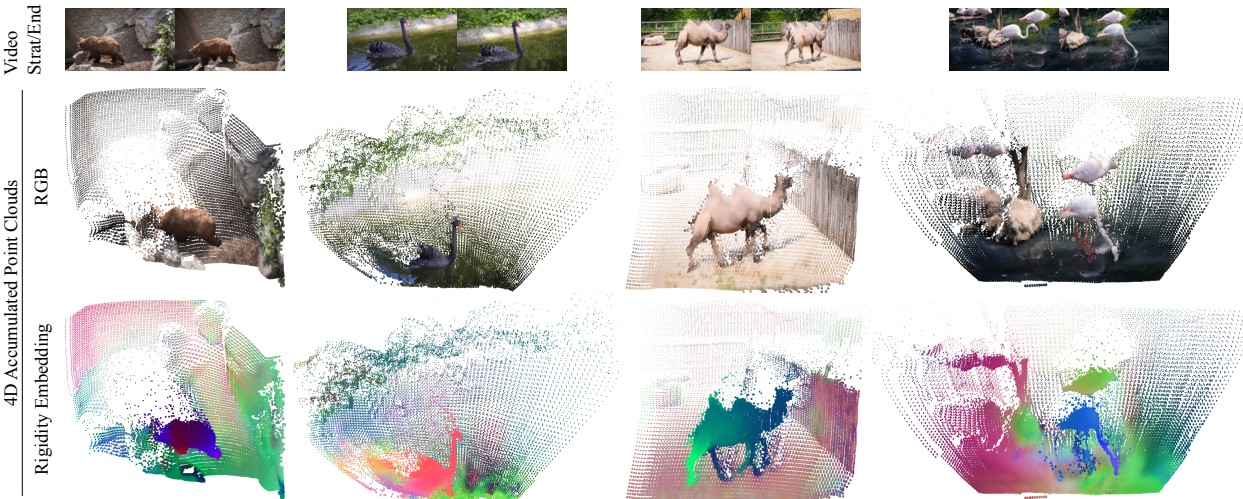

Figure 6: **Per-Scene Intermediate 4D Reconstructions Using Off-the-Shelf Depth.** While we emphasize that geometry estimation or 4D reconstruction is not the primary aim of our method, here we plot accumulated 4D point clouds and their corresponding unprojected rigidity embeddings after per-scene reconstruction using off-the-shelf monocular depth estimates.

coordinate frame and $\mathbf{P}_{i,t}$ maps camera-$t$ to camera-$(t+1)$, and motions between arbitrary times are obtained by chaining these relative poses from the start to end of the video sequence. Supervision is obtained by the SE(3)-induced reprojection loss of Eq. equation 3, which reprojects each point track from each timestep to each other timestep (dense re-projection in time) and compares projected 2D locations to the input 2D track locations.

## 4 Experiments and Results

### 4.1 Training SIRE

Our approach enables network optimization to be performed either per video, yielding a scene-specific network, or across a dataset of videos to train a generalizable prior. We demonstrate results in both settings to highlight that our method generates meaningful gradients and information even from a single video, and enables data-efficient training of priors over a dataset of videos. Results from Fig. 3 and Table 2 are optimized using our dataset-wide prior (though we perform a short additional fine-tuning step per scene to enhance the results in Fig. 3 on top of our feed-forward estimates), and results in Fig. 3, Fig. 6, and Table 1 are optimized from-scratch, per video.

While depth supervision can be incorporated when available, the model can also operate in a fully self-supervised manner, relying solely on 2D point tracks for supervision to learn priors over objects and geometry. This makes it particularly effective in real-world scenarios when explicit depth supervision is unavailable or expensive. As learning geometry from a single dynamic video is inherently ill-posed, our method can leverage depth supervision for more accurate per-scene reconstructions 4D estimations in this setting. Note that even in such challenging single-video cases without depth supervision, our 2D intrinsic rigidity embeddings are still compelling and useful. We only use ground-truth depth when explicitly indicated, such as in the 4D reconstructions of Fig. 6.

We leverage CO-Tracker (Karaev et al., 2023) for our 2D point track estimations, ML-PRO (Bochkovskii et al., 2024) and Video Depth Anything (Chen et al., 2025) for any ground-truth depth (we use Video Depth Anything's estimates scaled to be metric via ML-PRO), GeoCalib (Veicht et al., 2024) for estimates of intrinsics from each video, and Adam for optimization.

## 4.2   Memory and Time Requirements

Our approach is computationally efficient, with rapid convergence and modest memory requirements. When optimizing per scene (∼40 frames of a video), convergence is achieved within minutes, with just a few GB of VRAM. For full-scale training across multiple videos, we optimize on a single 48GB GPU, using a batch size of 120 frames (10-frame videos with batch size of 12). The model is trained for 30k iterations (∼0.5 days), though we observe that convergence is largely achieved within the first 5k iterations, demonstrating the compute- and data-efficiency of our method.

We evaluate SIRE's intrinsic rigidity embeddings and intermediate 4D reconstruction quality on multiple downstream tasks, including frozen feature segmentation estimation, self-supervised depth estimation, and SE(3) trajectory estimation (though we emphasize that these geometry reconstructions are intermediate byproducts of our method and not the primary objective). Our results suggest that our learned intrinsic rigidity embeddings contain semantically intuitive segmentation-like features of moving objects in videos and that our method can be used to learn powerful priors on large-scale video data. For results CO3D-Dogs, we pre-train our model on the entire CO3D-Dogs dataset, and for the DAVIS scenes, we fit our model from-scratch, per-video.

## 4.3   Downstream Feature Segmentation Training

To assess the quality of our learned rigidity embeddings, we freeze features from either our rigidity embeddings or an intermediate layer of an image backbone and train a two-layer MLP on top of the frozen features for object segmentation. That is, we train SIRE per-video, then freeze the backbone features or rigidity embeddings and separately train a small network on top of those frozen features in a downstream segmentation task. We evaluate on the DAVIS dataset (Caelles et al., 2019), which provides ground-truth moving object segmentations. We compare our features to those of the last layer of the backbone before training (referred to as "ImageNet Encoder"), the same layer of the backbone after training our model ("Our Encoder"), our rigidity embeddings ("Our Rig. Emb."), and features from DINOv2 (Oquab et al., 2023; Fu et al., 2024). See Fig. 5 for segmentation results.

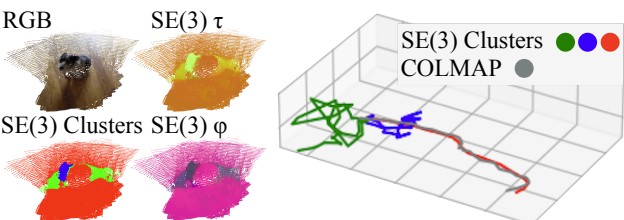

Figure 7: **SE(3) Trajectories** Though our method does not explicitly estimate camera poses, we cluster the per-track SE(3) estimates and compare the estimated trajectories with COLMAP estimated poses, since presumably one of our estimated clusters should correspond to camera motion.

In Table 1, we report IOU and observe that our method achieves stronger segmentations compared to the ImageNet-pretrained network (He et al., 2015). Our backbone features after training contains slightly sharper features than our rigidity embeddings, but using denser point tracks will likely reduce this gap. While DINO features are more informative than ours, note that our method was trained *per-video* here, whereas DINO was trained on large image datasets. On the same DAVIS scenes, we further highlight the soft 2D segmentations that our model produces as intermediate per-track rigidity maps in Fig. 4. We also demonstrate in Fig. 6 that, with depth supervision, our model can learn compelling 4D reconstructions even on single scenes.

## 4.4   Self-Supervised Depth Estimation

We evaluate self-supervised depth estimation on the CO3D-Dogs (Sinha et al., 2023) dataset, comparing our model against model ablations and FlowMap. We use ML-PRO (Bochkovskii et al., 2024) as ground-truth depth labels to compare our model against since the dataset does not provide ground-truth depth.

For the CO3D-Dogs dataset, we also mask our estimated rigidity weights with rigidity estimates from thresholded Sampson distance on off-the-shelf optical flow, as used in prior works (Zhang et al., 2024; Ye et al., 2022). The Sampson distance effectively measures how well an optical flow vector is explained by the RANSAC-estimated median SE(3) transformation (likely the camera pose), and the error is thresholded

such that only highly-confident vectors are marked as static. During training, we set the rigidity weights to be constant (completely rigid) if a point is within the rigid mask, and the predicted value otherwise. We find that using these masks further constrains the learned geometry, but is not necessary when using off-the-shelf depth estimates. See the supplement for more details.

In Table 2, we quantitatively compare our depth estimates with those from a FlowMap-trained model, and the following ablations of our model: a version without CO3D-Hydrants-pretraining, and a version without epipolar rigidity masks. See Fig. 3 for examples of learned depth maps.

To train our model, we use the 1k videos from the CO3D-Dogs dataset. For evaluating on depth estimation (above), we evaluate on the first frame of 100 videos, and for SE(3) trajectory estimation (below), we similarly evaluate trajectories on the first 30 frames of 100 videos.

### 4.5   SE(3) Trajectory Estimation

Our method does not explicitly estimate camera poses separately; it estimates SE(3) trajectories per-point-track. To evaluate our SE(3) with the dataset-provided COLMAP poses, we cluster the per-track SE(3) trajectories and report the minimum-loss trajectory from the clusters, assuming one corresponds to the camera motion. We report quantitative comparison (ATE) to COLMAP in Table 2 and visualization in Fig. 7. We compare with the same model ablations and ablations as above for depth estimation, and also additionally compare with MegaSaM (Li et al., 2024b), a recent camera pose estimation method for dynamic scenes. Both our model and MegaSaM both estimate poses in real-time after pre-processing (point tracks in our case and depth maps in theirs). Note that we evaluate MegaSaM just on poses because the ground-truth depth for evaluation comes from the same source that MegaSaM uses as input to its system. The fact that our poses are similar to COLMAP aligns with prior results from FlowMap, which has demonstrated COLMAP-comparable reconstructions, since our method reduces to FlowMap within each softly estimated rigid grouping.

| Method | Depth (MSE) | Poses (ATE) |
|---|---|---|
| Full | 0.6407 | 0.0044 |
| No Pretrain | 0.6441 | 0.0047 |
| No Rig. Mask | 0.7666 | 0.0088 |
| FlowMap | 0.8709 | 0.0466 |
| MegaSaM | — | 0.0104 |

Table 2: **Generalizable Depth and Pose Estimation Results.** We compare feedforward depth and SE(3) trajectory estimates from our method and baselines to those from ML-PRO (Bochkovskii et al., 2024) and COLMAP (Schönberger & Frahm, 2016).

## 5   Conclusion

We have introduced SIRE, a simple and differentiable formulation for learning priors over object rigidity from monocular video. Specifically, we show how to train intrinsic rigidity embeddings via a simple SE(3) reconstruction objective, self-supervised from monocular video and off-the-shelf point tracks. Through experiments we demonstrate that the intrinsic rigidity embeddings our model learns contain meaningful semantic information with high data-efficiency, paving the way towards learning priors over objects and rigid motion from either small-scale or large-scale video data. Note that our method indeed has limitations: our method depends on accurate point tracks, which are known to struggle on fluids and sky regions, as well as intrinsics, and can struggle to learn accurate geometry on scenes with insufficient or excessive motion.

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
