# OpenReview forum: "SIRE: SE(3) Intrinsic Rigidity Embeddings"
_TMLR — Accepted by TMLR_

### Review · Reviewer_ucXD · 2025-08-28

**Summary Of Contributions:**

In this paper is proposed a self supervised method to learn depth and 3d motion of point tracks, focussing on learning rigid body embeddings. In other works [Flowcam], [Flowmap], [RAFT-3D], the concept is generally featured - but differs in the flavour. As such, the method uses the same ingredients as Flowmap and Flowcam, but adds a incremental improvement in creating 'rigidity' embeddings, which forms the subject of this paper.

The pipeline is very similar to Flowmap:
1. Get matching points across video frames
2. Predict and lift to 3d.
3. Solve for relative pose using orthogonal procrustates (SVD)
4. Project to 2d (lets say from cam 1 to cam 2) and minimize flow loss available from flow solver (cotracker).

Rigidity embeddings are used to weight the orthogonal procrustates algorithm, and are learnt implicitly. It differs from RAFT in that we use scene flow labels to supervise it, and it is learnt without labels as such.

The authors show that using rigidity embeddings help
 - motion segmentation: comparison is made with DINOv2
- depth and pose are better than with the Flowmap

These embeddings are usable in downstream tasks as a useful supervision cue.

Flowcam: https://arxiv.org/pdf/2306.00180
Flowmap: https://arxiv.org/pdf/2404.15259v1
RAFT-3D: https://arxiv.org/pdf/2012.00726

**Audience:**

Yes

**Audience Explanation:**

This is a core computer vision and machine learning work. The TMLR audience will very much be interested.

**Claims And Evidence:**

Yes

**Claims Explanation:**

I like the general idea of learning interpretable embeddings that can help as object cues. It might be useful in tasks such as object detection and occupancy prediction too (I work in 3d vision for autonomous driving).

The solidity of the method is apparent. We build upon Flownet, but extend to learn object embeddings.

**Requested Changes:**

Kindly elaborate on how the embeddings are used in a mathematical formulation. I would generally the entire algorithm to be depicted mathematically. I am also interesting in learning how well this method would perform in outdoor scenes (e.g. highway). Can you run an experiment?

---

> ### Author Response · Authors · 2025-10-07
> **Pointer to Relevant Changes based on Feedback**
>
> We appreciate your interest in interpretability and deployment settings.
>
> • Full forward pass notation and descriptions (Main Sec. 3.5). We added a compact description of the full SIRE forward pass, covering per-frame prediction of depth/rigidity, the SE(3) track solve, chaining of per-track SE(3) motions, and the SE(3)-induced dense-in-time re-projection loss, also clarifying the frame of reference. The source of intrinsics are described in the experiments section (Main 4.1).
>
> • Architecture Illustration and Description (Supp. Sec. 1). We illustrated the network's predictions in the forward pass and detailed the pre-training or initialization of each component.
>
> • Additional multi-object datasets (Supp. Sec. 2). We ran SIRE on two more multi-object datasets: one of robot gripper demonstration videos and another on highway driving scenes. We visualize predicted rigidity embeddings and geometry, highlighting that our model can predict meaningful rigid object segmentations, especially on the robot dataset, but can struggle to segment multiple moving bodies when they exhibit parallel motion (cars dataset).

---

### Review · Reviewer_tFRj · 2025-09-20

**Summary Of Contributions:**

This paper introduces SIRE, a self-supervised framework that learns rigidity-aware embeddings from monocular video and off-the-shelf 2D point tracks. The method groups pixels into rigid components via embeddings and estimates per-track SE(3) trajectories, supervised by a simple reprojection loss. The key contributions are:
	•	A fully differentiable and simple formulation for learning intrinsic rigidity embeddings, enabling both dataset-wide training and per-video optimization.
	•	Extending prior FlowMap-style approaches from static scenes to dynamic videos, by solving per-track rather than global SE(3) motions.
	•	Demonstrating that rigidity embeddings capture semantically meaningful motion segmentation cues and are useful for downstream tasks such as segmentation, depth estimation, and trajectory estimation.
	•	Empirical validation on CO3D-Dogs and DAVIS datasets, showing competitive performance to baselines despite operating in a self-supervised and data-efficient setting.

Strengths:
	•	Conceptually simple and elegant framework; fully differentiable.
	•	Strong efficiency and data efficiency, converging quickly per-video.
	•	Produces embeddings with clear semantic association.

Weaknesses:
	•	Heavy reliance on point track quality; struggles likely under occlusion, fluids, or textureless regions.
	•	Evaluation scope is somewhat narrow, focused on CO3D-Dogs and DAVIS, with limited demonstrations on multi-object dynamic scenes.
	•	Missing implementation details, including architecture/initialization of the depth head and embedding head.
	•	Requires per-video optimization during inference, limiting practicality compared to fully feed-forward alternatives.

**Audience:**

Yes

**Audience Explanation:**

The paper focuses on self-supervised learning, motion segmentation, and 3D scene understanding, all highly relevant to the TMLR topics. The proposed approach is simple, efficient, and general enough to influence research on self-supervised dynamic 3D learning from large-scale videos. Many in the audience will be interested in the idea of learning rigidity priors without explicit supervision.

**Broader Impact Concerns:**

No major ethical concerns arise from this work.

**Claims And Evidence:**

Yes

**Claims Explanation:**

The authors provide convincing evidence through both qualitative and quantitative experiments. The segmentation results demonstrate that rigidity embeddings capture motion-based structure. The depth and pose experiments validate the correctness of the SE(3) results. The ablation studies (e.g., without pretraining, without rigidity masks) support the value of the rigidity embeddings learning.

Some aspects could be better substantiated, such as performance in more complex dynamic multi-object scenes and robustness under occlusion, but the presented evidence sufficiently supports the main claims.

The evaluation in Section 4.3 is a bit concerning. SIRE is trained per-video (closer to an overfitting setup), while the baselines such as ImageNet Encoder and DINOv2 are generic pretrained representations. Although the authors note this difference, the comparison still risks overstating SIRE’s advantage.

**Requested Changes:**

- Clarify implementation details, including architecture/initialization of the depth head and embedding head. (P0)
- Clarify the frame of reference for SE(3) trajectories (camera-centered, first frame, or another convention). (P0)
- Expand evaluation to more diverse datasets with multiple independently moving objects (e.g., 100DoH, BONN, Stereo4D). Current experiments are limited in scope and may not stress-test the method’s generality. (P1)
- Discuss robustness under occlusion and track dropouts. Since point tracks are the backbone of the approach, the paper should address how the method behaves when tracks are noisy. It would also be useful to provide qualitative examples of failure cases (e.g., scenes with fluids, textureless regions, or large occlusions). (P2)
- Clarify why MegaSaM is evaluated only on poses and not depth. (P2)

---

> ### Author Response · Authors · 2025-10-07
> **Pointer to Relevant Changes based on Feedback**
>
> We appreciate your careful reading and detailed suggestions!
>
>
> • Architecture Illustration and Description (Supp. Sec. 1). We illustrated the network's predictions in the forward pass and detailed the pre-training or initialization of each component.
>
> • Full forward pass notation and descriptions (Main Sec. 3.5). We added a compact description of the full SIRE forward pass, covering per-frame prediction of depth/rigidity, the SE(3) track solve, chaining of per-track SE(3) motions, and the SE(3)-induced dense-in-time re-projection loss, also clarifying the frame of reference.
>
> • Additional multi-object datasets (Supp. Sec. 2). We ran SIRE on two more multi-object datasets: one of robot gripper demonstration videos and another on highway driving scenes. We visualize predicted rigidity embeddings and geometry, highlighting that our model can predict meaningful rigid object segmentations, especially on the robot dataset, but can struggle to segment multiple moving bodies when they exhibit parallel motion (cars dataset).
>
> • Ablations: Robustness under bad tracks, depth supervision, and embedding dimensions (Supp. Sec. 3, 1.1). We ran SIRE on a video sequence where a duck floats over water, inducing noisy and lost point tracks (illustrated). We show that without depth supervision in this challenging case and when trained just on this single video (no dataset-wide priors), our model can struggle to coherently group rigid bodies, and that when adding depth supervision, the rigid bodies become intuitively and coherently grouped. We also ablate and describe the choice of rigidity embedding dimension one the CO3D-Dogs dataset and explain this hyper parameter has a 'sweet spot' between over and under parameterized dimensions.
>
> • MegaSaM clarification: We clarify (Main 4.5) why we only compare it on poses.

---

> > ### Comment · Reviewer_tFRj · 2025-10-27
> > **Final recommendation**
> >
> > Thank you for the response and detailed revisions. The additional experiments and clarifications improve the clarity and completeness of the paper. However, I am a little bit confused by the results on the multi-object dataset. It would be helpful to include more explanation about the evaluation setup. If the model is trained per-video, the fact that it separates the gripper and the object into distinct embeddings even when there is no relative motion between them (as shown in the provided frames) suggests that SIRE may still be learning instance-level segmentation masks rather than capturing true intrinsic rigidity. Similarly, the results on the driving dataset appear blurry and do not convincingly demonstrate motion-based grouping. It would also be informative to contrast these results against DINO features or other motion-segmentation baselines.
> >
> > Btw, there's a new paper in ICCV'25 looks related RoMo: Robust Motion Segmentation Improves Structure from Motion https://romosfm.github.io/
> >
> > Overall, I find the core idea of SIRE both elegant and promising. The paper presents convincing qualitative and quantitative evidence in several sections, but the new multi-object results do not yet clearly isolate learned rigidity from standard instance segmentation. Strengthening the evaluation through clearer experimental design or comparing with baselines would significantly enhance the paper.
> >
> > I lean towards acceptance. The conceptual contribution is interesting and potentially impactful, but the current multi-object experiments are limited and the results are not yet clearly useful for downstream tasks.

---

### Review · Reviewer_KKcA · 2025-09-24

**Summary Of Contributions:**

This work introduces SIRE (SE(3) Intrinsic Rigidity Embeddings), which is a self-supervised video learning framework that learns pixel-wise rigidity embeddings and depth maps, which are then used to solve for and estimate local per-track SE(3) transformations. The key contribution lies in the use of rigidity embeddings to softly group points into rigid components, allowing the model to capture multiple independently moving rigid objects without supervision, as well as the loss function based on SE(3) induced 2D projection, which enables self-supervised training without the need for ground truth labels for trajectories, rigidity or depth.

Strengths:
- The model has a simple but effective training formulation, using an end-to-end differentiable pipeline based on reprojection error, which help avoids dependence on labels and heavy supervision.
- The framework seems to be very data efficient, it shows good results both in dataset-wide and per-video optimization regimes, demonstrating flexibility.
- Despite this apparent simplicity, the framework achieves competitive results on segmentation, depth, and pose estimation tasks.

Weaknesses:
- The paper provides almost no details about the network backbone, layers, or design choices, relying heavily on references for these details. I had trouble figuring out whether the embedding networks were CNNs or some other type of layer, as this is only mentioned in passing in one figure caption, and there are no further details provided about the layer. This harms the paper severely,  as it makes it difficult to understand for readers outside the narrow field and makes the work much more difficult to reproduce.
- Similarly to the previous point, equations are introduced without properly explaining variables and which of them are learnable or not, and what their learning setup is in case they are learnable.
- The ablation presented in the work is minimal, with only a few cases being shown (“No Pretrain,” “No Rig. Mask”) , making it unclear which other design choices contributed to the performance, such as embedding dimensionality, network depth, or supervision choices, etc.
- The scope of the evaluations is fairly narrow, with benchmarks being restricted to DAVIS and CO3D-Dogs. More diverse datasets would better validate generality.

**Audience:**

Yes

**Audience Explanation:**

The paper would be of interest to TMLR readers working in:
- Self-supervised representation learning
- 3D scene understanding and structure-from-motion
- Video segmentation and motion analysis

The idea of intrinsic rigidity embeddings is appealing as a reusable prior, and the self-supervised nature is well in-line with current trends in large-scale video learning. That said, the paper could be more attractive if it provided, greater clarity on the architecture and implementation details, as it is currently really difficult for readers outside of the scene reconstruction community to understand. There is no mention of layer types, widths or depths for any almost part of the model. Furthermore, in equations such as the Procrustes formulation, some of the variables such as K are never explained. I assume this represents the camera intrinsics matrix, but it is entirely unclear whether this information is obtained as part of the datasets or also needs to be estimated with the model, and if it needs to be estimated no details are provided on how this is done.

So, while the audience may likely find it interesting, readability and impact of the paper is currently hampered by a lack in many technical details which needs to be improved with revisions.

**Broader Impact Concerns:**

No broader impact concerns.

**Claims And Evidence:**

Yes

**Claims Explanation:**

In general, yes, the core claims are backed by quantitative comparisons, qualitative visualizations (rigidity maps, point clouds), and ablations that show the importance of induced rigidity embeddings and the flexible training procedure.

However, the narrow scope of datasets makes it unclear whether the method generalizes beyond the tested settings. Evaluation on some other established datasets that include video data (e.g., LLFF, MipNeRF-360 among others) will make the results much more convincing.

Additionally, the lack of detailed ablations makes it difficult to assess which parts of the pipeline are actually contributing to the improved performance. This also makes it harder to guess whether the framework will be useful for other scenarios that were not explored in the paper.

Overall the evidence is convincing but incomplete: strong enough to support the paper’s immediate claims, but not detailed enough to determine the wider applicability of the approach.

**Requested Changes:**

1. Add architectural details (critical)
    - Explicitly describe the encoder backbone, with basic details such as network type (CNN/ResNet/UNet), layers, embedding dimensionality, depth. An in depth architecture description with a graphical overview needs to be at least provided as a part of the supplement.
     - Clarify exaclty how rigidity embeddings and depth maps are predicted, and also provide more details on how the camera intrinsics matrix is obtained. Do the embedding layers share weigths, are they independent, and other details relevant to reproducibility.

2. Provide code (critical)
   - I couldn't find any code or repository linked in the paper or as a part of the supplementary materials. This is essential for any ML paper to be published.

3. Expand ablations (would strengthen work)
   - Study embedding dimensionality, different encoder types, or training with/without depth supervision.

4. Broader evaluations (would strengthen work)
    - Test and compare on more diverse benchmarks (e.g., LLFF, MipNeRF-360 among others).

5. Address typographical issues (critical)
   - Page 5, last paragraph: errors in the Procrustes equation formatting, some variables contain commas between * and t, and others do not (e.g. p*t and p*, t+1)
   - Page 6, last paragraph: " two points are in same rigid body if they similar embeddings" seems like a 'have' is missing there.
   - A careful proofreading pass is recommended for other typos that may have been missed.

---

> ### Author Response · Authors · 2025-10-07
> **Pointer to Relevant Changes based on Feedback**
>
> Thank you for the thoughtful feedback and clear pointers.
>
> • Architecture Illustration and Description (Supp. Sec. 1). We illustrated the network's predictions in the forward pass and detailed the pre-training or initialization of each component.
>
> • Ablations: Robustness under bad tracks, depth supervision, and embedding dimensions (Supp. Sec. 3, 1.1). We ran SIRE on a video sequence where a duck floats over water, inducing noisy and lost point tracks (illustrated). We show that without depth supervision in this challenging case and when trained just on this single video (no dataset-wide priors), our model can struggle to coherently group rigid bodies, and that when adding depth supervision, the rigid bodies become intuitively and coherently grouped. We also ablate and describe the choice of rigidity embedding dimension one the CO3D-Dogs dataset and explain this hyper parameter has a 'sweet spot' between over and under parameterized dimensions.
>
> • Additional multi-object datasets (Supp. Sec. 2). We ran SIRE on two more multi-object datasets: one of robot gripper demonstration videos and another on highway driving scenes. We visualize predicted rigidity embeddings and geometry, highlighting that our model can predict meaningful rigid object segmentations, especially on the robot dataset, but can struggle to segment multiple moving bodies when they exhibit parallel motion (cars dataset).
>
> • Code release. We will publicly release code upon acceptance to further benefit the vision community.
>
> • Typos & notation cleanup. We fixed observed typos and took a thorough pass over the text to increase clarity in notation.
>
>
> • Full forward pass notation and descriptions (Main Sec. 3.5). We added a compact description of the full SIRE forward pass, covering per-frame prediction of depth/rigidity, the SE(3) track solve, chaining of per-track SE(3) motions, and the SE(3)-induced dense-in-time re-projection loss, also clarifying the frame of reference. The source of intrinsics are described in the experiments section (Main, 4.1).

---

### Decision · Action_Editor_bgfA · 2025-11-09

**Recommendation:** Accept as is

**Audience:**

Yes

**Audience Explanation:**

The paper deals with learning 3D aware representations and thus touches on multiple topics of clear interest to the community.

**Claims And Evidence:**

Yes

**Claims Explanation:**

The revision clarified a number of points about the paper and improved the manuscript. In the end, all three expert reviewers were in favor of accepting the paper. The AE has examined the reviews and the paper and agrees with the reviewer consensus.